# MARCH8 inhibits viral infection by two different mechanisms

**Yanzhao Zhang[1†], Takuya Tada[1†‡], Seiya Ozono[1,2], Satoshi Kishigami[2], Hideaki Fujita[3], Kenzo Tokunaga[1]\***

[1]Department of Pathology, National Institute of Infectious Diseases, Tokyo, Japan; [2]Faculty of Life and Environmental Sciences, University of Yamanashi, Yamanashi, Japan; [3]Faculty of Pharmaceutical Sciences, Nagasaki International University, Nagasaki, Japan

**Abstract** Membrane-associated RING-CH 8 (MARCH8) inhibits infection with both HIV-1 and vesicular stomatitis virus G-glycoprotein (VSV-G)-pseudotyped viruses by reducing virion incorporation of envelope glycoproteins. The molecular mechanisms by which MARCH8 targets envelope glycoproteins remain unknown. Here, we show two different mechanisms by which MARCH8 inhibits viral infection. Viruses pseudotyped with the VSV-G mutant, in which cytoplasmic lysine residues were mutated, were insensitive to the inhibitory effect of MARCH8, whereas those with a similar lysine mutant of HIV-1 Env remained sensitive to it. Indeed, the wild-type VSV-G, but not its lysine mutant, was ubiquitinated by MARCH8. Furthermore, the MARCH8 mutant, which had a disrupted cytoplasmic tyrosine motif that is critical for intracellular protein sorting, did not inhibit HIV-1 Env-mediated infection, while it still impaired infection by VSV-G-pseudotyped viruses. Overall, we conclude that MARCH8 reduces viral infectivity by downregulating envelope glycoproteins through two different mechanisms mediated by a ubiquitination-dependent or tyrosine motif-dependent pathway.

**\*For correspondence:**
tokunaga@nih.go.jp

[†]These authors contributed equally to this work

**Present address:** [‡]Department of Microbiology, New York University School of Medicine, New York, United States

**Competing interests:** The authors declare that no competing interests exist.

## Introduction

Membrane-associated RING-CH (MARCH) 8 is one of 11 members of the MARCH family of RING-finger E3 ubiquitin ligases, which consist of an N-terminal cytoplasmic tail (CT) domain containing a C4HC3 RING finger (RING-CH finger) motif, two transmembrane (TM) domains, between which a short ectodomain is located, and a C-terminal CT domain (*Bartee et al., 2004*; *Goto et al., 2003*). MARCH8 downregulates a variety of cellular transmembrane proteins, such as MHC-II (*Ohmura-Hoshino et al., 2006*), CD86 (*Tze et al., 2011*), CD81 (*Bartee et al., 2010*), CD44 (*Eyster et al., 2011*), TRAIL receptor 1 (*van de Kooij et al., 2013*), CD98 (*Eyster et al., 2011*), IL-1 receptor accessory protein (*Chen et al., 2012*), and transferrin receptor (*Fujita et al., 2013*). We have recently reported that MARCH8 reduces HIV-1 infectivity by downregulating HIV-1 envelope glycoproteins (Env) from the cell surface, resulting in a reduced incorporation of Env into virions (*Tada et al., 2015*). Intriguingly, vesicular stomatitis virus G-glycoprotein (VSV-G) was even more sensitive to the inhibitory effect of MARCH8. In the case of HIV-1 Env, it is retained intracellularly without degradation after cell-surface downregulation. In contrast, VSV-G is not only downregulated from the cell surface but also undergoes lysosomal degradation by MARCH8 (*Tada et al., 2015*). In this regard, we hypothesized that VSV-G, whose cytoplasmic tail is lysine-rich (5 out of 29 amino acids), could be readily ubiquitinated by the E3 ubiquitin ligase MARCH8 and therefore undergo lysosomal degradation, whereas HIV-1 Env carries only two lysines (out of 151 amino acids) in its cytoplasmic tail and may rarely undergo degradation after being trapped by MARCH8. In this study, we created lysine mutants of both HIV-1 Env and VSV-G, together with newly generated MARCH8 mutants to explore

the hypothesis described above. The results with these mutants show that MARCH8 targets HIV-1 Env and VSV-G by two different inhibitory mechanisms.

## Results and discussion

We have recently reported that MARCH8 inhibits lentiviral infection by reducing virion incorporation of both HIV-1 Env and VSV-G in a RING-CH domain-dependent manner. Because the RING-CH domain is known to be essential for the E3 ubiquitin ligase activity of MARCH8, we asked whether these envelope glycoproteins are susceptible to MARCH8-mediated ubiquitination. To investigate this, we first created the VSV-G mutant CT5K/R in which five arginine residues were introduced in place of cytoplasmic lysine residues that could be ubiquitination targets (*Figure 1A*, upper). We also generated the HIV-1 Env gp41 mutant CT2K/R harboring two arginines in place of the cytoplasmic lysines (*Figure 1A*, lower). Then, we prepared HIV-1 luciferase reporter viruses pseudotyped with the mutant envelope glycoproteins (VSV-G CT5K/R and HIV-1 Env CT2K/R) from 293T cells transiently expressing MARCH8, and compared their viral infectivity with that of control viruses pseudotyped with wild-type (WT) envelope glycoproteins. The infectivity of viruses harboring either VSV-G CT5K/R or HIV-1 gp41 Env CT2K/R was almost comparable to WT-enveloped viruses (*Figure 1B and C*). As expected, the virus pseudotyped with VSV-G CT5K/R was completely resistant to MARCH8 (*Figure 1B*). In contrast, the HIV-1 Env CT2K/R-pseudotyped virus was still susceptible to the inhibitory effect of MARCH8 (*Figure 1C*). Consistent with these results, immunofluorescence staining showed that the five lysine mutations in the CT domain of VSV-G conferred resistance to MARCH8-mediated intracellular degradation (*Figure 1D*), whereas the two lysine mutations in the CT domain of HIV-1 Env had no effect on its cell-surface downregulation by MARCH8 (*Figure 1E*). It should be noted that MARCH8-resistant VSV-G CT5K/R colocalized with MARCH8 (*Figure 1D*). We thus speculated that MARCH8 ubiquitinates lysine residues of VSV-G but not of HIV-1 Env at their CT domains. Importantly, in the presence of MARCH8, HIV-1 Env preferentially colocalized with the trans-Golgi network (TGN) marker TGN46 (*Figure 1F*), consistent with our previous hypothesis that MARCH8 might sequester HIV-1 Env in the TGN without degradation (*Tada et al., 2015*).

To investigate this possibility, we performed immunoprecipitation (IP)/western-based ubiquitination assays. In cells coexpressing WT HA-MARCH8, VSV-G was efficiently ubiquitinated, whereas in cells coexpressing the RING-CH mutant of HA-MARCH8, the ubiquitination of VSV-G was lost. More importantly, the VSV-G lysine mutant CT5K/R did not undergo MARCH8-mediated ubiquitination, as expected (*Figure 1G*), suggesting that the lysine residues at the CT domain of VSV-G are specifically ubiquitinated by MARCH8. These findings are consistent with the immunofluorescence results (*Figure 1D and E*). We therefore conclude that lysine residues at the CT domain of VSV-G are ubiquitinated by MARCH8, which determines the difference in the MARCH8-mediated intracellular fate between these viral glycoproteins.

Because unlike VSV-G, HIV-1 Env was retained intracellularly without degradation, as we previously reported (*Tada et al., 2015*), we hypothesized that these viral envelope glycoproteins might undergo endocytosis with different mechanisms of action. It has been reported that another MARCH family member, MARCH11, has a conserved tyrosine-based motif, YXXϕ, which is known to be recognized by the adaptor protein (AP) μ-subunits in the C-terminal CT domain (*Morokuma et al., 2007*). We therefore looked for the same motif(s) in MARCH8 and found the $^{222}$YxxL$^{225}$ and $^{232}$YxxV$^{235}$ sequences in the CT domain at the C-terminus (*Figure 2A*). Based on this finding, we generated tyrosine motif mutants of MARCH8, in which either $^{222}$Y or $^{232}$Y was mutated to alanine (designated $^{222}$AxxL$^{225}$ or $^{232}$AxxV$^{235}$). The protein expression in cells transfected with each MARCH8 plasmid was confirmed by immunoblotting using an anti-hemagglutinin (HA) antibody (*Figure 2B*). Then, we examined whether these YXXϕ motifs are important for the antiviral activity of MARCH8. The infectivity of VSV-G–pseudotyped viruses was still inhibited by the expression of both $^{222}$AxxL$^{225}$ and $^{232}$AxxV$^{235}$ MARCH8 mutants (*Figure 2C*). In contrast, the infectivity of HIV-1 Env–pseudotyped viruses was not impaired by $^{222}$AxxL$^{225}$ MARCH8 expression but was reduced by that of $^{232}$AxxV$^{235}$ and WT MARCH8 (*Figure 2D*), suggesting that the first tyrosine motif ($^{222}$YxxL$^{225}$) is involved in the antiviral activity of MARCH8 against HIV-1 Env but not VSV-G.

We previously reported that the MARCH8-mediated reduction in viral infectivity was due to reduced entry efficiency, resulting from a decreased virion incorporation of envelope glycoproteins through their downregulation from the cell surface (*Tada et al., 2015*). Therefore, by performing a

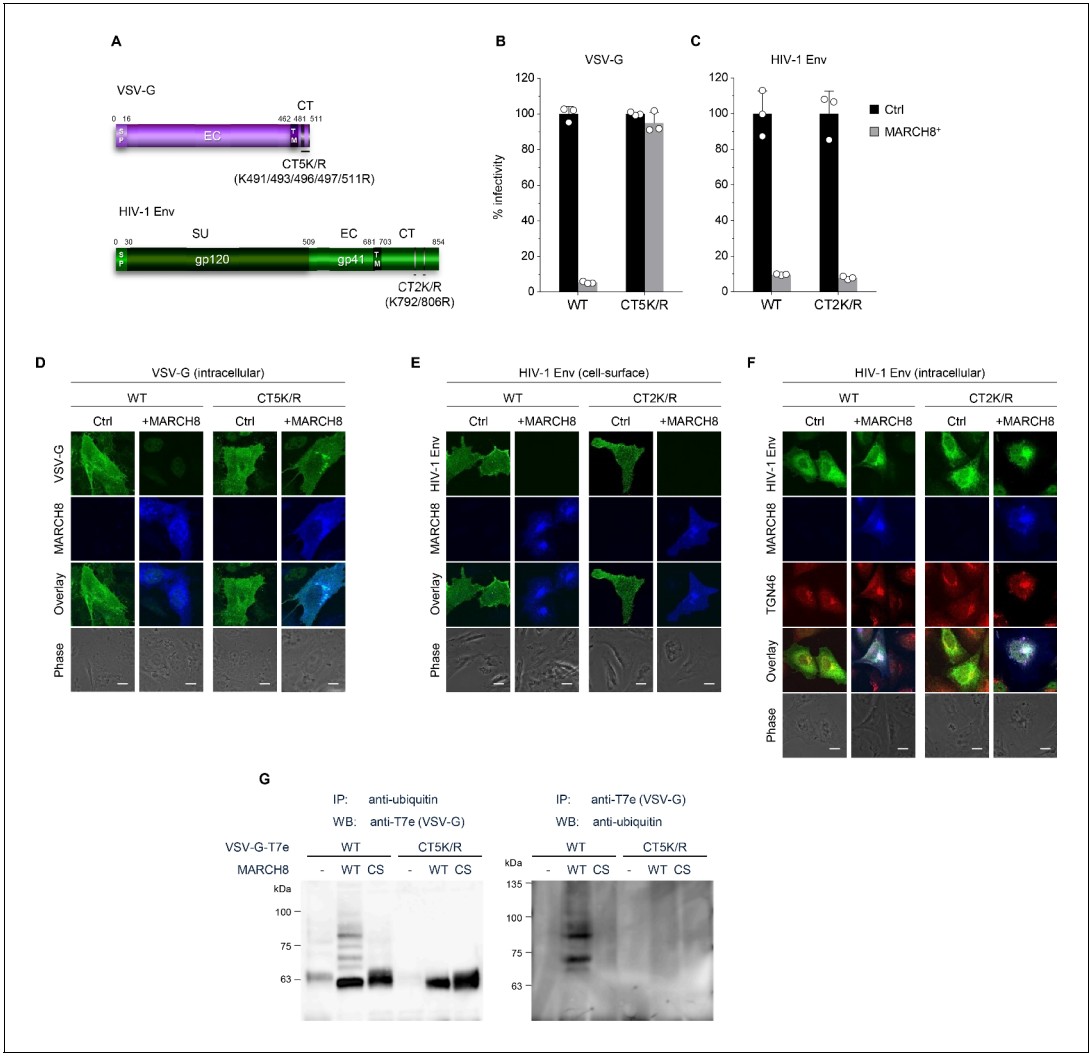

**Figure 1.** MARCH8 targets and ubiquitinates cytoplasmic lysine residues of VSV-G but not of HIV-1 Env. (**A**) Schematic structure of the lysine mutants of VSV-G (CT5K/R; upper) and HIV-1 Env (CT2K/R; lower). SP, signal peptide; EC, extracellular domain; TM, transmembrane domain; CT, cytoplasmic tail; SU, surface subunit. (**B**) Infectivity of viruses prepared from 293T cells cotransfected with Env-defective HIV-1 luciferase (luc) reporter proviral DNA and either the VSV-G wild-type (WT) or CT5K/R mutant plasmid together with either a control (Ctrl) (black) or HA-MARCH8 (gray) plasmid. Data are shown as a percentage of the viral infectivity in the absence of MARCH8 when WT VSV-G was used (mean + s.d. from three independent experiments). (**C**) Infectivity of viruses prepared as shown in *B*, except for using either the WT HIV-1 Env or its CT2K/R mutant plasmid (mean + s.d. from three independent experiments). (**D**) The VSV-G lysine mutant is resistant to MARCH8-mediated intracellular degradation. Shown are immunofluorescence-based analyses of the intracellular expression of either the WT or CT5K/R mutant VSV-G with or without MARCH8 in transfected HOS cells. Scale bars, 10 μm. Note that the cell-staining for VSV-G cannot be performed because VSV-G is tagged with the T7 epitope (T7e) at the C-terminus. (**E**) The lysine mutant of HIV-1 Env is still sensitive to MARCH8-induced downregulation from the cell surface. Immunofluorescence images show cell-surface expression of either the WT or CT2K/R mutant HIV-1 Env with or without MARCH8 in transfected HOS cells. Scale bars, 10 μm. (**F**) MARCH8 sequesters HIV-1 Env in the trans-Golgi network (TGN). Immunofluorescence images show intracellular localization of either the WT or CT2K/R mutant HIV-1 Env and the TGN marker TGN46, with or without MARCH8 in transfected HOS cells. Scale bars, 10 μm. (**G**) Lysine residues at the CT domain of VSV-G are ubiquitinated by MARCH8. The ubiquitination of the WT or CT5K/R mutant VSV-G tagged with T7e in cells expressing control or MARCH8 (WT or RING-CH mutant (CS)) was examined by immunoprecipitation (IP) of either ubiquitinated proteins with an anti-ubiquitin antibody (left panel) or of T7e-tagged VSV-G with an anti-T7e antibody (right panel), followed by immunoblotting with an antibody to either T7e (left panel) or ubiquitin (right panel), respectively.

β-lactamase (BlaM)-fused viral protein R (Vpr)-based entry assay, we first focused on whether the loss of function of $^{222}$AxxL$^{225}$ MARCH8 against HIV-1 Env but not VSV-G would indeed be due to the loss of its inhibitory activity on viral entry. Whereas the entry of VSV-G–pseudotyped HIV-1 prepared from cells expressing either WT or $^{222}$AxxL$^{225}$ MARCH8 was reduced compared with that of

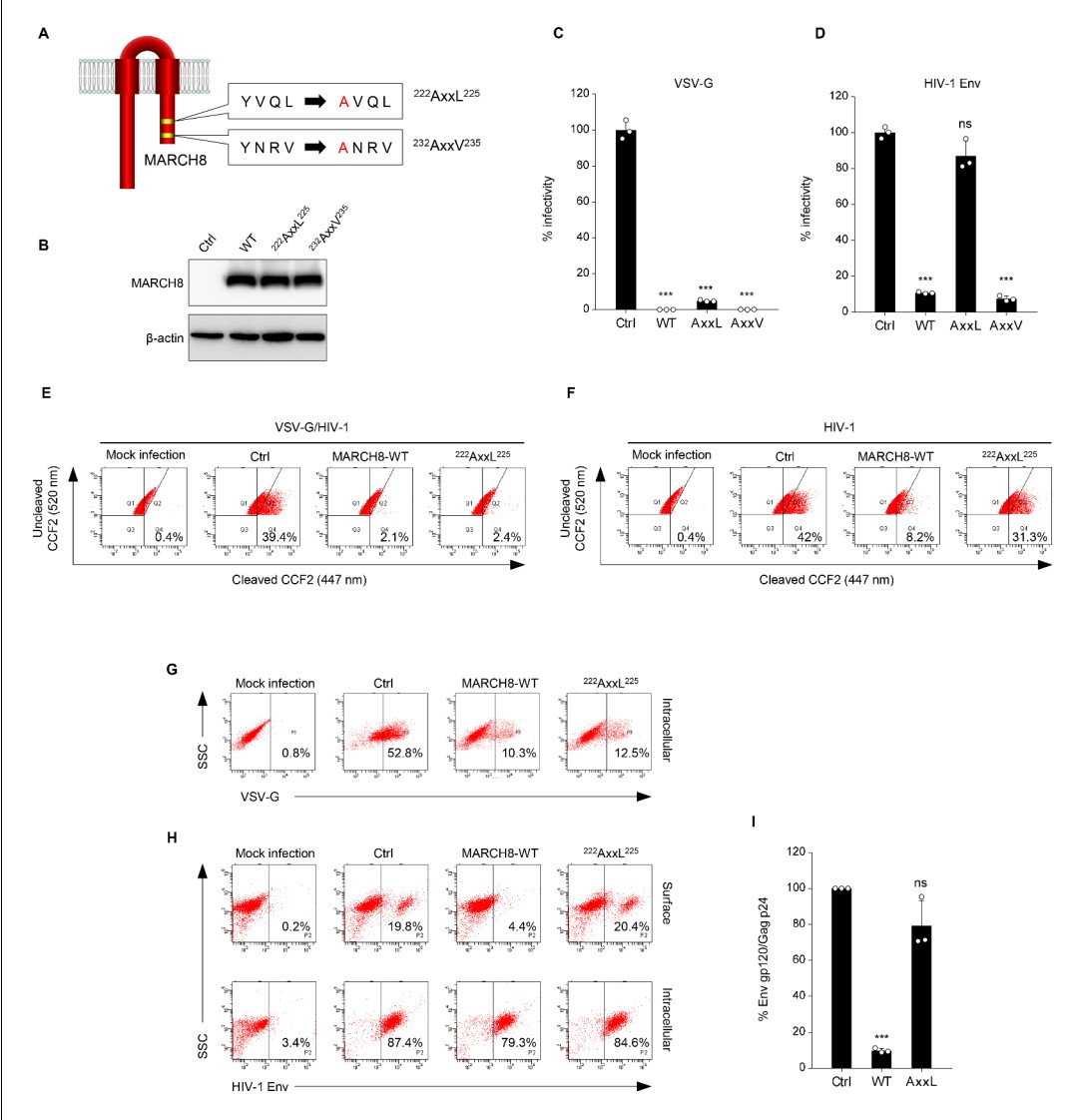

**Figure 2.** The tyrosine motif of MARCH8 mediates downregulation of HIV-1 Env but not of VSV-G. (A) Schematic structure of YxxΦ motif mutants of MARCH8 ($^{222}$YxxL$^{225}$ and $^{232}$YxxV$^{235}$). (B) Western blot analysis was performed by using extracts from 293T cells transfected with HA-tagged MARCH8 expression plasmids. Antibodies specific for HA were used to detect MARCH8 proteins. (C, D) Infectivity of viruses prepared from 293T cells cotransfected with Env-defective HIV-1 luciferase (luc) reporter proviral DNA and either a control (Ctrl), HA- WT, HA-$^{222}$AxxL$^{225}$ or HA-$^{232}$AxxV$^{235}$ MARCH8 plasmid, together with either (C) the VSV-G expression plasmid or (D) the HIV-1 Env expression plasmid. Data are shown as a percentage of the viral infectivity in the absence of MARCH8 (mean + s.d. from three independent experiments). ns; ***p<0.0005 compared with the Ctrl using two-tailed unpaired *t*-tests. (E, F) BlaM-Vpr-based viral entry assay using VSV-G-pseudotyped viruses (E) or NL4-3 whole viruses (F) produced from cells expressing either control, WT MARCH8, or the $^{222}$AxxL$^{225}$ mutant. Representative FACS dot plots are shown from four independent experiments. (G) VSV-G is downregulated by both WT and $^{222}$AxxL$^{225}$ mutant MARCH8, (H) whereas the cell-surface expression of HIV-1 Env is not affected by the mutant MARCH8. (I) $^{222}$AxxL$^{225}$ MARCH8 expression in producer cells is unable to decrease HIV-1 gp120 levels in viral supernatants. ELISA-based levels of Env gp120 in viral supernatants from 293T cells cotransfected with luc reporter proviral DNA and NL-Env plasmid, together with either MARCH8 WT or its $^{222}$AxxL$^{225}$ mutant. Representative data from three independent experiments are shown as percent Env gp120/Gag p24 in the supernatants relative to that from control cells. (mean + s.d. from three independent experiments). ns; ***p<0.0005 compared with the Ctrl using two-tailed unpaired *t*-tests.

the control virus (*Figure 2E*), the inhibition of the entry of whole HIV-1 virions was abrogated in viruses produced from cells expressing $^{222}$AxxL$^{225}$, as expected (*Figure 2F*). We further analyzed whether this motif of MARCH8 is indeed involved in the reduced virion incorporation of HIV-1 Env, which results from its cell-surface downregulation. To address this, we conducted flow cytometric analysis and quantified the levels of cell-surface and intracellular expression of Env glycoproteins. In

accordance with the results obtained in infectivity assays (*Figure 2C*), $^{222}$AxxL$^{225}$ MARCH8 still reduced intracellular VSV-G expression as well as WT MARCH8 did (*Figure 2G*). On the other hand, the mutant MARCH8 had a completely abrogated ability to downregulate cell-surface HIV-1 Env, whereas WT expression led to the downregulation of HIV-1 Env from the cell surface and its intracellular retention (*Figure 2H*), as we previously observed (*Zhang et al., 2019*). The results were consistent with those of the inhibitory activity of MARCH8 against the virion incorporation of HIV-1 Env (*Figure 2I*). We therefore conclude that $^{222}$YxxL$^{225}$ is critical for the MARCH8-mediated downregulation of HIV-1 Env but not VSV-G, which results in its reduced virion incorporation leading to impaired viral entry.

In summary, we first show the two different mechanisms by which MARCH8 inhibits viral infections, one being a ubiquitin-dependent downregulation that mediates lysosomal degradation of VSV-G whose cytoplasmic lysine residues are recognized by the RING-CH domain of MARCH8 (*Figure 3*, left), and the other, a YxxΦ motif-dependent downregulation that could explain the intracellular retention of HIV-1 Env in the TGN without degradation after cell-surface downregulation (*Figure 3*, right). In terms of the latter mechanism, although this could be attributed to the AP-dependent trafficking that requires the YxxΦ motif, which binds to AP μ-subunits, our preliminary results were unable to show the interaction between these subunits and MARCH8, due to technical difficulties in detecting specific bands in our IP/western blotting assays. We have recently reported that MARCH1 and MARCH2 are also antiviral MARCH family members that inhibit HIV-1 infection, although their antiviral activity is less robust, which is probably due to the lower protein expression and/or stability than that of MARCH8 (*Zhang et al., 2019*). Because MARCH1 and MARCH2 also harbor the Yxxφ motif in their C-terminal CT domains, it would be intriguing to verify the importance of the motif in these proteins. It should be noted that in this study, we tested either HIV-1 Env or VSV-G derived from a single representative strain, which is a limitation of our present findings. Nevertheless, the present findings are consistent with our previous studies showing that MARCH8-induced downregulation of VSV-G leads to lysosomal degradation, while that of HIV-1 Env results in intracellular retention without degradation. In this context, the reason why VSV infection in human is always subclinical (*Hanson et al., 1950*) could be explained at least in part by the observation that VSV-G is completely inactivated even by lower levels of MARCH8 (*Tada et al., 2015*). Conversely, in the case of HIV-1 infection, it is intriguing to postulate that the virus might rather take advantage of infection of macrophages enriched with MARCH8 (*Tada et al., 2015*; *Zhang et al., 2019*) as an ideal hideout, in which the virus could escape from immunosurveillance potentially due to the MARCH8-induced downregulation of Env from the cell surface. Overall, we conclude that MARCH8 targets HIV-1 Env and VSV-G by two different inhibitory mechanisms (either ubiquitin-dependent or YxxΦ

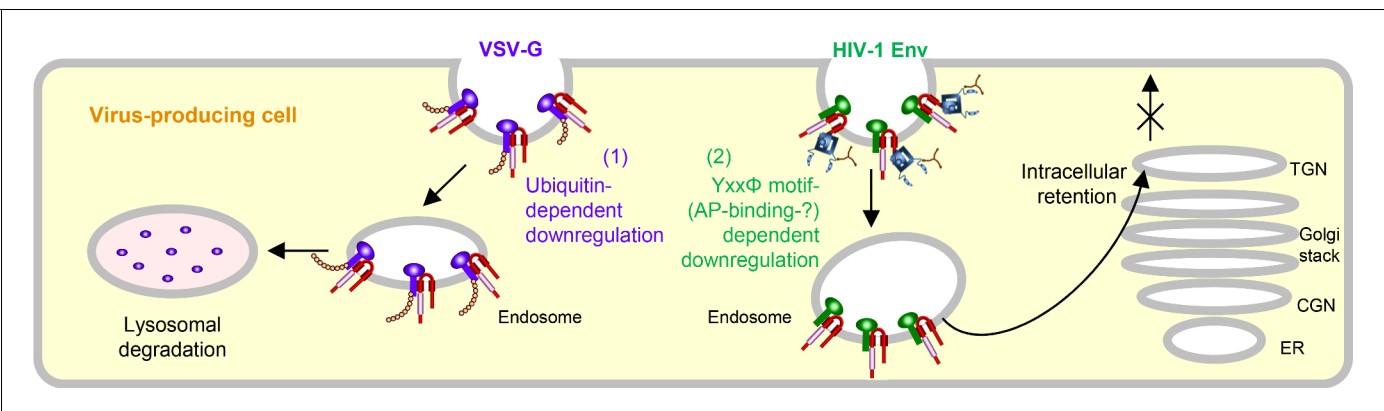

**Figure 3.** Schematic diagram of two different molecular mechanisms by which MARCH8 inhibits viral infection. *Left*, MARCH8 (red) downregulates VSV-G (violet) in a ubiquitin-dependent manner. The RING-CH domain (pink) of MARCH8 recognizes VSV-G's cytoplasmic lysine residues, which results in ubiquitin conjugation (shown as orange beads), leading to lysosomal degradation; *Right*, MARCH8 downregulates HIV-1 Env (green) in a YxxΦ motif-dependent manner. The tyrosine motif located in the C-terminal CT of MARCH8 likely interacts with the adaptor protein μ-subunits (navy) (if this is the case with μ2 or μ1, clathrin (brown) is involved in this step), resulting in the intracellular retention of HIV-1 Env in the TGN without degradation. It should be noted that the downregulation of these viral glycoproteins might not necessarily occur at the plasma membrane. The nucleus and other organelles are not shown.

motif-dependent downregulation). Further investigations will clarify the more detailed host defense mechanisms of this protein.

## Materials and methods

### DNA constructs

The Env-deficient HIV-1 proviral indicator construct pNL-Luc2-E(-), HIV-1 Gag-Pol expression plasmid pC-GagPol-RRE, HIV-1 Env-expression vector pC-NLenv, HIV-1 Rev expression plasmid pCa-Rev, VSV-G expression plasmid pC-VSVg and its C-terminally T7-epitope-tagged version pVSVg-T7E, GFP expression plasmid pCa-EGFP, Vpr/β-lactamase (BlaM) expression plasmid pMM310, and MARCH8 expression plasmid either pC-MARCH8 or pC-HA-MARCH8 and its RING-CH mutant pC-HA-MARCH8-CS have previously been described elsewhere (*Iwabu et al., 2009*; *Tada et al., 2015*). The HIV-1 Env gp41 mutant CT2K/R, in which cytoplasmic lysine residues at positions 792 and 806 were mutated to arginine, and the VSV-G mutant CT5K/R, in which cytoplasmic lysine residues at positions 491, 493, 496, 497, and 511 were mutated to arginine residues, were created by inserting overlapping PCR fragments into *Mfe*I/*Xho*I-digested pC-NLenv and by inserting PCR fragments into *Kpn*I/*Not*I-digested pC-VSVg (or pVSVg-T7E), respectively. The MARCH8 mutant $^{222}$AxxL$^{225}$ or $^{232}$AxxV$^{235}$, in which a tyrosine residue at position 222 or 232 was mutated to an alanine residue, was generated by inserting overlapping PCR fragments into the *Kpn*I/*Xho*I-digested pCAGGS or *Xho*I/*Not*I-digested pCAGGS-NHA to create an untagged or N-terminally HA-tagged expression plasmid. All constructs were verified by a DNA sequencing service (FASMAC).

### Cell maintenance

293T, MT4 (RRID:CVCL_2632), HeLa, MAGIC5 (HeLa derivative [*Mochizuki et al., 1999*]), and HOS cells were maintained under standard conditions. Cells were originally obtained from ATCC (except MAGIC5 cells) and routinely tested negative for mycoplasma contamination (PCR Mycoplasma Detection kit, Takara).

### Viral infectivity assays

To prepare VSV-G-pseudotyped or HIV-1-Env-pseudotyped luciferase reporter viruses, $2.5 \times 10^5$ 293T cells were cotransfected with 120 ng of either the MARCH expression plasmid (note that this DNA amount was previously optimized to reflect endogenous expression in macrophages) or a control plasmid, 20 ng of pC-VSVg, pC-VSVg-CT5K/R, pC-NLenv, or pC-NLenv-CT2K/R, 500 ng of pNL-Luc2-E(-), and an empty vector up to 1 µg of total DNA, using FuGENE 6 (Promega). Sixteen hours later, the cells were washed with phosphate-buffered saline (PBS), and 1 ml of fresh complete medium was added. After 24 hr, supernatants were treated as described above and then harvested. The p24 antigen levels in viral supernatants were measured by an HIV-1 p24 antigen capture ELISA (XpressBio). To determine viral infectivity, $1 \times 10^4$ MAGIC5 cells were incubated with 1 ng of p24 antigen from the HIV-1 supernatants. After 48 hr, cells were lysed in 100 µl of One-Glo Luciferase Assay Reagent (Promega). The firefly luciferase activity was determined with a Centro LB960 (Berthold) luminometer.

### Immunoblotting assays

Protein expression of constructs was confirmed by western blot analyses as described elsewhere (*Tada et al., 2015*; *Zhang et al., 2019*). Briefly, cells transfected as described above were lysed in 500 µl of lysis buffer containing 1.25% n-octyl-β-D-glucoside (Dojindo), and Complete protease inhibitor cocktail (Roche Applied Science). Cell extracts were then subjected to gel electrophoresis and transferred to a nitrocellulose membrane, followed by probing with an anti-HA mouse monoclonal antibody (Sigma-Aldrich, H9658) or an anti-β-actin mouse monoclonal antibody (Sigma-Aldrich, A5316). Proteins were then visualized by chemiluminescence using an ECL western blotting detection system (GE Healthcare) and monitored by using a LAS-3000 imaging system (FujiFilm).

### Viral entry assays

A β-lactamase (BlaM)-fused viral protein R (Vpr)-based entry assay was performed as described elsewhere (*Tada et al., 2015*; *Zhang et al., 2019*). Briefly, HIV-1 particles containing a fusion protein of

BlaM-Vpr were produced by cotransfection of 293 T cells (5 X $10^5$) with 1 µg of pNL4-3 (*Adachi et al., 1986*), 300 ng of pMM310 (*Tobiume et al., 2003*) encoding BlaM-Vpr, 240 ng of either the control vector, pC-MARCH8, pC-MARCH8-$^{222}$AxxL$^{225}$ or pC-MARCH8-$^{232}$AxxV$^{235}$, and the control vector up to 1 µg of total DNA. Similarly, VSV-G-pseudotyped HIV-1 particles containing BlaM-Vpr were prepared by cotransfection with 1 µg of pNL-E(-) (*Iwabu et al., 2009*), 40 ng of pC-VSVg, 300 ng of pMM310, 240 ng of either the control vector, pC-MARCH8, pC-MARCH8-$^{222}$-AxxL$^{225}$, pC-MARCH8-$^{232}$AxxV$^{235}$, and or the control vector up to 1 µg of total DNA. The produced viruses were normalized to the p24 antigen level (100 ng) and used for infection of the CD4$^+$ T cell line MT4 (5 × $10^5$ cells) at 37°C for 4 hr to allow viral entry. After extensive washing with Hank's balanced salt solution (HBSS; Invitrogen), cells were incubated with 1 µM CCF2-AM dye (Invitrogen), a fluorescent substrate of BlaM, in HBSS containing 1 mg ml$^{-1}$ Pluronic F-127 surfactant (Invitrogen) and 0.001% acetic acid for 1 hr at room temperature and then washed with HBSS. Cells were further incubated for 14 hr at room temperature in HBSS supplemented with 10% FBS, washed three times with PBS and fixed in a 1.2% paraformaldehyde solution. Fluorescence was monitored at 520 and 447 nm by flow cytometry using BD FACS Canto II (BD Bioscience), and the data were collected and analyzed with BD FACS Diva Software (BD Bioscience).

## Flow cytometry

293T cells (5 × $10^5$) were cotransfected with 0.8 µg of either the control vector and 200 ng of pCa-EGFP, or the combination of 120 ng of either the MARCH expression plasmid (WT or a tyrosine mutant) or the control vector, 20 ng of either pC-VSVg or pC-NLenv, 500 ng of pNL-Luc2-E(-), and the control vector up to 1 µg. To analyze cell-surface expression, the transfected cells were incubated with either anti-HIV-1 gp120 goat polyclonal antibody (Abcam, ab21179) or anti-VSV-G mouse monoclonal antibody (Sigma-Aldrich, V5507), followed by staining for 30 min on ice with either a rabbit anti-goat IgG conjugated with Alexa 647 (Molecular Probes, A21446) or a goat anti-mouse IgG conjugated with R-phycoerythrin (Molecular Probes P-852), respectively. Cells were washed extensively with PBS with 4% FBS and fixed with 4% formaldehyde in PBS. To analyze intracellular expression, the transfected cells were fixed with 0.01% formaldehyde in PBS, permeabilized with 0.05% saponin for 10 min, and immunostained with the antibody against either gp120 or VSV-G, followed by incubation with secondary antibodies, as described above. GFP-positive cells were sorted and analyzed for the expression of HIV-1 Env or VSV-G by flow cytometry using a BD FACS Canto II, and the data were collected and analyzed with BD FACS Diva Software.

## Env incorporation assays

HIV-1 gp120 ELISA-based Env incorporation assays were performed by using an HIV-1 gp120 antigen capture ELISA kit (Advanced BioScience Laboratories), according to the manufacturer's instructions. To normalize Env levels in viral supernatants, p24 antigen was measured by the HIV-1 p24 antigen capture ELISA.

## Ubiquitination assays

293T cells (5 × $10^5$) were cotransfected with 0.8 µg of pC-VSVg-T7E, 0.4 µg of the control vector, and 0.8 µg of either the control vector, pC-HA-MARCH8, or pC-HA-MARCH8-CS. After 48 hr, cells were lysed in TBS-T buffer (50 mM Tris-HCl buffer (pH 7.5), 0.15 M NaCl, 1% Triton X-100, and 0.5% deoxycholic acid) containing a protease inhibitor cocktail and 10 mM N-ethylmaleimide, as an inhibitor of deubiquitination enzymes. The mixture was centrifuged at 21,500 × g for 15 min, and the supernatant was used as total cell lysate for immunoblotting or IP. Fifty microliters of Protein A-coupled Sepharose 4B (GE Healthcare, 17-0780-01) was preincubated for 2 hr at 4°C with 4 µg of the appropriate antibody (anti-T7 epitope rabbit polyclonal antibody, MBL, PM022; anti-ubiquitin mouse monoclonal antibody Clone FK2, Cayman, 14220). Total cell lysate was incubated with antibody-coupled Sepharose for 20 hr at 4°C. The Sepharose was washed three times with TBS-T buffer and one time with PBS before the immunoprecipitated proteins were eluted with SDS sample buffer. To evaluate the ubiquitination states of the immunoprecipitated proteins, proteins immunoprecipitated with an anti-T7 epitope rabbit antibody were subjected to western blotting with an anti-ubiquitin mouse antibody, whereas proteins immunoprecipitated with the anti-ubiquitin mouse antibody were subjected to western blotting with the anti-T7 epitope rabbit antibody. Immunoreactive bands were

detected using an ECL detection kit (EzWestLumi plus, ATTO) with a ChemiDoc imaging system (Bio-Rad).

## Immunofluorescence microscopy

HOS cells were plated on 13 mm coverslips, cotransfected with the indicated plasmids, 0.5 μg of either pC-NLenv (WT or CT2K/R) or pC-VSVg-T7E (WT or CT5K/R), 0.1 μg of pC-Gag-Pol, 0.05 μg of pCa-Rev, and 0.3 μg of either pC-HA-MARCH8 expression plasmid or a control plasmid using FuGENE6 and cultured for 24 hr. For the total staining of VSV-G and MARCH8, cells were fixed with 4% paraformaldehyde for 30 min on ice and permeabilized with 0.05% saponin (Sigma-Aldrich). The fixed cells were incubated with primary antibodies, anti-T7 epitope mouse monoclonal antibody (Novagen, 69522–4) and anti-HA goat polyclonal antibody (GenScript, A00168-40). The secondary antibodies, Alexa 488 donkey anti-mouse IgG (Molecular Probes, A-21202) and Alexa 647 donkey anti-goat IgG (Molecular Probes, A-21447) were used for the double staining assay. For the cell sur-face staining of gp120 protein, cells were incubated with an anti-gp120 mouse monoclonal antibody (0.5β Matsushita et al., 1988; kindly provided by S. Matsushita) at 4°C for 5 min and washed with PBS at 4°C before fixation. Fixation was performed with 4% paraformaldehyde for 30 min on ice, and fixed cells were permeabilized with 0.05% saponin to detect the intracellular expression and localiza-tion of MARCH8 proteins. The coverslips were incubated with the anti-HA rabbit polyclonal antibody (Sigma Aldrich, H6908) for 1 hr, washed with PBS and incubated for 30 min with the secondary anti-bodies Alexa 488 donkey anti-mouse IgG (Molecular Probes, A-21202) and Alexa 647 donkey anti-rabbit IgG (Molecular Probes, A-31573) for the double staining assay. For the total staining of gp120, MARCH8, and TGN46 (as a trans-Golgi network (TGN) marker), the fixed cells were incu-bated with primary antibodies, anti-HIV-1 gp120 goat polyclonal antibody (Abcam, ab21179), anti-HA mouse monoclonal antibody (Sigma Aldrich, H3663), and anti-TGN46 rabbit polyclonal antibody (Abcam, ab50595) for 1 hr, washed with PBS and followed by the incubation for 30 min with the sec-ondary antibodies, Alexa 488 donkey anti-goat IgG (Molecular Probes, A-11055), Alexa 568 donkey anti-rabbit IgG (Molecular Probes, A-10042) and Alexa 647 donkey anti-mouse IgG (Molecular Probes, A-31571) for the triple staining. Confocal images were obtained with a FluoView FV10i auto-mated confocal laser-scanning microscope (Olympus; Tokyo).

## Statistical analyses

Column graphs that combine bars and individual data points were created with GraphPad Prism ver-sion 8.04. *P*-values generated from two-tailed unpaired *t*-tests for data represented in *Figures 2C, D and I*.

## Acknowledgements

We thank S Matsushita (Kumamoto University, Japan) for the anti-gp120 mouse monoclonal anti-body 0.5β. This work was supported by grants from the Japan Society for the Promotion of Science (18K07156) to KT.

## Additional information

### Funding

| Funder | Grant reference number | Author |
| --- | --- | --- |
| Japan Society for the Promo-tion of Science | 18K07156 | Kenzo Tokunaga |

The funders had no role in study design, data collection and interpretation, or the decision to submit the work for publication.

### Author contributions

Yanzhao Zhang, Takuya Tada, Data curation, Formal analysis, Validation, Investigation; Seiya Ozono, Formal analysis, Investigation; Satoshi Kishigami, Resources; Hideaki Fujita, Data curation, Formal analysis, Validation, Investigation, Visualization, Methodology; Kenzo Tokunaga, Conceptualization,

Resources, Data curation, Software, Formal analysis, Supervision, Funding acquisition, Validation, Investigation, Visualization, Methodology, Writing - original draft, Project administration, Writing - review and editing

**Author ORCIDs**
Kenzo Tokunaga (iD) https://orcid.org/0000-0002-2625-5322

**Decision letter and Author response**
Decision letter https://doi.org/10.7554/eLife.57763.sa1
Author response https://doi.org/10.7554/eLife.57763.sa2

## Additional files
### Supplementary files
• Source data 1. Source Data File for *Figures 1B, C*, *2C, D and I*.

• Source data 2. Source data for *Figure 1G* and *2B*. Original uncropped images of IP-western blot (ubiquitination assays) in *Figure 1G*. The PVDF membranes were incubated with an anti-T7-epitope tag antibody, or with an anti-ubiquitin antibody. Images shown in *Figure 1G* were cropped from the boxed areas, and the brightness/contrast was adjusted equally across the entire image using Photoshop CS6 *Figure 2B* source data. Original uncropped images of western blot in *Figure 2B*. The PVDF membrane was incubated with an anti-HA antibody, then stripped and reprobed with an anti-β-actin antibody for a loading control. Images shown in *Figure 2B* were cropped from the boxed areas, and the brightness/contrast was adjusted equally across the entire image using Photoshop CS6.

• Transparent reporting form

### Data availability
All data generated or analyzed during this study are included in the manuscript.

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

# Appendix 1

**Appendix 1—key resources table**

| Reagent type (species) or resource | Designation | Source or reference | Identifiers | Additional information |
|---|---|---|---|---|
| Gene (*Homo sapiens*) | MARCH8 | NCBI | BC025394 | |
| Recombinant DNA reagent | pNL4-3 | *Adachi et al., 1986* | | |
| Recombinant DNA reagent | pNL-E(-) | *Iwabu et al., 2009* | | |
| Recombinant DNA reagent | pNL-Luc2-E(-) | *Tada et al., 2015* | | |
| Recombinant DNA reagent | pC-GagPol-RRE | *Tada et al., 2015* | | |
| Recombinant DNA reagent | pC-NLenv | *Tada et al., 2015* | | |
| Recombinant DNA reagent | pCa-Rev | *Iwabu et al., 2009* | | |
| Recombinant DNA reagent | pC-VSVg | *Tada et al., 2015* | | |
| Recombinant DNA reagent | pVSVg-T7E | *Tada et al., 2015* | | |
| Recombinant DNA reagent | pCa-EGFP | *Iwabu et al., 2009* | | |
| Recombinant DNA reagent | pMM310 | *Tobiume et al., 2003* | | |
| Recombinant DNA reagent | pC-MARCH8 | *Tada et al., 2015* | | |
| Recombinant DNA reagent | pC-HA-MARCH8 | *Tada et al., 2015* | | |
| Recombinant DNA reagent | pC-HA-MARCH8-CS | *Tada et al., 2015* | | |
| Recombinant DNA reagent | pC-NLenv-CT2K/R | This paper | | See Materials and methods |
| Recombinant DNA reagent | pC-VSVg-CT5K/R | This paper | | See Materials and methods |
| Recombinant DNA reagent | pC-VSVg-CT5K/R-T7E | This paper | | See Materials and methods |
| Recombinant DNA reagent | pC-MARCH8-$^{222}$AxxL$^{225}$ | This paper | | See Materials and methods |
| Recombinant DNA reagent | pC-HA-MARCH8-$^{222}$AxxL$^{225}$ | This paper | | See Materials and methods |
| Recombinant DNA reagent | pC-MARCH8-$^{232}$AxxV$^{235}$ | This paper | | See Materials and methods |
| Recombinant DNA reagent | pC-HA-MARCH8-$^{232}$AxxV$^{235}$ | This paper | | See Materials and methods |
| Sequence-based reagent | NL-env-MfeI-S | This paper | gacaattggagaagtgaatt | Sense primer for pC-NLenv-CT2K/R |
| Sequence-based reagent | NLenv-CT2KR-A | This paper | ctattcC ttagttcctgactccaatactgt aggagattccaccaatatC tgagggc | Overlapping PCR's antisense primer for pC-NLenv-CT2K/R |

*Continued on next page*

*Appendix 1—key resources table continued*

| Reagent type (species) or resource | Designation | Source or reference | Identifiers | Additional information |
|---|---|---|---|---|
| Sequence-based reagent | NLenv-CT2/KR-S | This paper | gccctcaG atattggtggaatctcctaca gtattggagtcaggaactaaG gaatag | Overlapping PCR's sense primer for pC-NLenv-CT2K/R |
| Sequence-based reagent | NL-env-XhoI-A | This paper | ccgCTCGAG ttatagcaaaatc ctttccaag | Antisense primer for pC-NLenv-CT2K/R |
| Sequence-based reagent | VSVg-BsiWI-S | This paper | atCGTACG atgaagtgccttttgtactt | Sense primer for pC-VSVg-CT5K/R-T7E |
| Sequence-based reagent | VSVg-CT5K/R-XhoI-A | This paper | ATctcgaGcC ttccaagtcggttcatctc tatgtctgtataaatctgtcttC tcCtggt gtgcCttaatCtaatg | Antisense primer for pC-VSVg-CT5K/R-T7E |
| Sequence-based reagent | MARCH8-KpnI-S | This paper | ggGGTACC atgagcatgccact gcatcag | Sense primer for pC-MARCH8-222AxxL225 and pC-MARCH8-232AxxV235 |
| Sequence-based reagent | MARCH8-XhoI-S | This paper | ccgCTCGAG agcatgccactg catcagat | Sense primer for pC-HA-MARCH8-222AxxL225 and pC-HA-MARCH8-232AxxV235 |
| Sequence-based reagent | MARCH8-222AxxL225-S | This paper | gtgtaaagtgGCtgtgcaG ttgt ggaagag | Overlapping PCR's sense primer for pC-MARCH8-222AxxL225 and pC-HA-MARCH8-222AxxL225 |
| Sequence-based reagent | MARCH8-222AxxL225-A | This paper | ctcttccacaaCtgcacaGC cact ttacac | Overlapping PCR's antisense primer for pC-MARCH8-222AxxL225 and pC-HA-MARCH8-222AxxL225 |
| Sequence-based reagent | MARCH8-232AxxV235-S | This paper | gagactcaaggccGC taataga gtgatc | Overlapping PCR's sense primer for pC-MARCH8-232AxxV235 |
| Sequence-based reagent | MARCH8-232AxxV235-A | This paper | gatcactctattaGC ggccttg agtctc | Overlapping PCR's antiense primer for pC-MARCH8-232AxxV235 |
| Sequence-based reagent | MARCH8-XhoI-A | This paper | ccgCTCGAG tcagacgtgaatg atttctg | Antisense primer for pC-MARCH8-222AxxL225 and pC-MARCH8-232AxxV235 |
| Sequence-based reagent | MARCH8-NotI-A | This paper | attGCGGCCGC tcagacgtga atgatttctg | Antisense primer for pC-HA-MARCH8-222AxxL225 and pC-HA-MARCH8-232AxxV235 |
| Cell line (*H. sapiens*) | 293T | ATCC | CRL-3216 | |
| Cell line (*H. sapiens*) | MT4 | JCRB | 1216 RRID:CVCL_2632 | |
| Cell line (*H. sapiens*) | HeLa | ATCC | CVCL_0030 | |
| Cell line (*H. sapiens*) | MAGIC5 | *Mochizuki et al., 1999* | | |
| Cell line (*H. sapiens*) | HOS | ATCC | CRL-1543 | |
| Commercial assay or kit | PCR Mycoplasma Detection Set | Takara | TKR-6601 | Mycoplasma detection |

*Continued on next page*

*Appendix 1—key resources table continued*

| Reagent type (species) or resource | Designation | Source or reference | Identifiers | Additional information |
|---|---|---|---|---|
| Chemical compound, drug | FuGENE6 | Promega | E2691 | Transfection reagent |
| Commercial assay or kit | HIV-1 p24 ELISA Kit | XpressBio | XBR-1000 | HIV-1 p24 antigen capture ELISA |
| Commercial assay or kit | HIV-1 gp120 ELISA Kit | Advanced BioScience Laboratories | 5429 | HIV-1 gp120 ELISA |
| Commercial assay or kit | One-Glo Luciferase Assay Reagent | Promega | E6110 | Luciferase assay |
| Chemical compound, drug | Protein A-Sepharose | GE Healthcare | 17-0780-01 | Immunoprecipitation |
| Chemical compound, drug | Complete protease inhibitor cocktail | Roche | 11697498001 | Protease inhibitor |
| Chemical compound, drug | n-octyl-β-D-glucoside | Dojindo | O001 | Nonionic surfactant |
| Chemical compound, drug | Saponin | Sigma-Aldrich | 47036 | Nonionic surfactant |
| Antibody | Anti-HA | Sigma-Aldrich | H9658 RRID:AB_260092 | WB (1:10,000 ) Mouse monoclonal |
| Antibody | Anti-HA | Sigma-Aldrich | H3663 RRID:AB_262051 | IF (1:200) Mouse monoclonal |
| Antibody | Anti-HA | Sigma-Aldrich | H6908 RRID:AB_260070 | IF (1:200) Rabbit polyclonal |
| Antibody | anti-HA | GenScript | A00168-40 | IF (1:200) Goat polyclonal |
| Antibody | Anti-β-actin | Sigma-Aldrich | A5316 RRID:AB_476743 | WB (1:5,000) Mouse monoclonal |
| Antibody | Anti-T7 epitope tag | MBL | PM022 RRID:AB_592788 | IP (4 µg); WB (1:1,000) Rabbit polyclonal |
| Antibody | Anti-T7 epitope tag | Novagen | 69522-4 RRID:AB_11211744 | IF (1:200) Mouse monoclonal |
| Antibody | Anti-ubiquitin | Cayman | 14220 | IP (4 µg); WB (1:500) Mouse monoclonal (Clone FK2) |
| Antibody | Anti-gp120 | Abcam | Ab21179 RRID:AB_732949 | FACS (1:150); IF (1:200) Goat polyclonal |
| Antibody | Anti-gp120 | *Matsushita et al., 1988* | | IF (1:100) Mouse monoclonal (0.5β) kindly provided by S. Matsushita |
| Antibody | Anti-TGN46 | Abcam | Ab50595 RRID:AB_2203289 | IF (1:200) Rabbit polyclonal |
| Antibody | Anti-VSV-G | Sigma-Aldrich | V5507 RRID:AB_261877 | FACS (1:150) Mouse monoclonal |

*Continued on next page*

*Appendix 1—key resources table continued*

| Reagent type (species) or resource | Designation | Source or reference | Identifiers | Additional information |
|---|---|---|---|---|
| Antibody | Goat anti-mouse IgG conjugated with R-phycoerythrin | Molecular Probes | P-852 RRID:AB_143191 | FACS (1:500) |
| Antibody | Alexa 488 donkey anti-mouse IgG | Molecular Probes | A-21202 RRID:AB_141607 | IF (1:400) |
| Antibody | Alexa 488 donkey anti-goat IgG | Molecular Probes | A-11055 RRID:AB_2534102 | IF (1:400) |
| Antibody | Alexa 568 donkey anti-rabbit IgG | Molecular Probes | A-10042 RRID:AB_2534017 | IF (1:400) |
| Antibody | Alexa 647 donkey anti-goat IgG | Molecular Probes | A-21447 RRID:AB_141844 | FACS (1:500); IF (1:400) |
| Antibody | Alexa 647 donkey anti-mouse IgG | Molecular Probes | A-31571 RRID:AB_162542 | IF (1:400) |
| Antibody | Alexa 647 donkey anti-rabbit IgG | Molecular Probes | A-31573 RRID:AB_2536183 | IF (1:400) |
| Commercial assay or kit | ECL Western blotting detection system | GE Healthcare | RPN2109 | Chemiluminescence |
| Commercial assay or kit | EzWestLumi plus | ATTO | WSE-7120 | Chemiluminescence |
| Chemical compound, drug | HBSS | Thermo Fisher | 14025076 | Wash buffer |
| Chemical compound, drug | CCF2-AM dye | Invitrogen | K1023 | Fluorescent substrate for BlaM |
| Chemical compound, drug | Pluronic F-127 | Invitrogen | P2443 | Nonionic surfactant for CCF2-AM dye |
| Chemical compound, drug | Saponin | Sigma-Aldrich | 47036 | Non-ionic surfactant for immunofluorescence |
| Software, algorithm | BD FACS Diva Software | BD Bioscience | | |
| Software, algorithm | GraphPad Prism 8.04 | GraphPad | | |

