## [Decision Letter]

**Acceptance summary:**

The study reports differences in the mechanism of antiviral activity of the cellular protein MARCH8 on two different viruses, HIV and VSV. The study demonstrates that while MARCH8 downregulates the envelope protein of HIV, it does not degrade this HIV protein. This is in contrast to VSV-G, where MARCH8 downregulates the viral envelope protein and degrades it. The study illustrates a differential mechanisms of antiviral inhibition for the same host protein with different viruses.

**Decision letter after peer review:**

Thank you for submitting your article "MARCH8 inhibits viral infection by two different mechanisms" for consideration by *eLife*. Your article has been reviewed by three peer reviewers, one of whom is a member of our Board of Reviewing Editors, and the evaluation has been overseen by Päivi Ojala as the Senior Editor. The reviewers have opted to remain anonymous.

The reviewers have discussed the reviews with one another and the Reviewing Editor has drafted this decision to help you prepare a revised submission.

This manuscript follows earlier findings (Tada et al., 2015) in which they identified MARCH8 as a novel antiviral factor that inhibited VSV-G and HIV-1 Envelope to differing degrees. In the present manuscript, the authors revisit the antiviral activity of MARCH8, which has been found to downregulate several cellular transmembrane proteins. In this study the authors show that MARCH8 reduces cell surface Envelope via two pathways: ubiquitination and internalization via tyrosine motif. They show through viral entry, ubiquitination, and immunofluorescence assays that MARCH8 inhibits VSV-G through ubiquitination of its lysine residues, whereas HIV-1 is inhibited by MARCH8 through a conserved cytoplasmic tyrosine motif critical for intracellular protein sorting. This study suggest that the conserved antiviral function of MARCH8 is mediated by different mechanisms for these two viruses.

Summary:

The study of Zhang et al. reports differences in the mechanism of MARCH8 antiviral effects on HIV versus VSV-G. The study follows from a 2015 Nature Medicine paper showing that MARCH8 inhibits HIV replication by reducing the levels of Env on the cell and virus surface. The present study extends this work to show that while MARCH8 downregulates HIV Env, it does not degrade ENV. This is in contrast to VSV-G, where MARCH8 downregulates the viral protein and degrades it. They show evidence that the viral envelope proteins might undergo endocytosis differently, one being ubiquitin dependent the other dependent on a tyrosine motif. The study illustrates a differential mechanisms of antiviral inhibition for the same host protein with different viruses.

Essential revisions:

The authors conclude that VSV-G and HIV Env are targeted by two mechanisms of MARCH8 that are substantially different. This conclusion would be strengthened and the impact of the manuscript would be increased if the authors provided a better mechanistic understanding of how March 8 affects the two viral proteins differently. The major conclusions do require a modest amount of additional new data to be fully supported. The major conclusion of this publication is that MARCH8 reduces cell-surface and virion incorporation of HIV-1 Envelope. In contrast, MARCH8 downregulates VSV glycoprotein from cell-surface and degrades it via ubiquitin-mediated degradation. The mechanistic details behind the fate of HIV-1 Envelope are lacking. Since the substantive claim of the manuscript is the distinctive mechanism(s) by which MARCH8 exhibits its antiviral effects on HIV-1 and VSV – it is necessary that at least the following comment be addressed:

The loss of cell surface localization of HIV-1 Envelope upon MARCH8 overexpression is striking. However, as presented, the details regarding intracellular retention of HIV-1 Env are not clear. This is an essential claim of this study and thus it is recommended that microscopy experiments be performed to elucidate this mechanism: where is HIV-1 Env being sorted? This can be easily done by performing colocalization staining with known intracellular compartment markers.

Overall, the mechanism by which MARCH-8 internalizes HIV Env is under-developed. The observation that the tyrosine at position 222 is required is interesting. It is tempting to speculate that one or more AP complexes are involved, but the authors state their preliminary data, which they do not show, does not support this hypothesis. Despite this, in their model they still speculate that AP complexes are involved. The contradictory statements are very confusing.

The authors also need to explain how the levels of MARCH8 expression levels in their system compare to endogenous levels in HIV target cells. Over expression studies are notorious for artifacts and if the levels are higher, this must be noted.

---

## [Author Response]

Essential revisions:The authors conclude that VSV-G and HIV Env are targeted by two mechanisms of MARCH8 that are substantially different. This conclusion would be strengthened and the impact of the manuscript would be increased if the authors provided a better mechanistic understanding of how March 8 affects the two viral proteins differently. The major conclusions do require a modest amount of additional new data to be fully supported. The major conclusion of this publication is that MARCH8 reduces cell-surface and virion incorporation of HIV-1 Envelope. In contrast, MARCH8 downregulates VSV glycoprotein from cell-surface and degrades it via ubiquitin-mediated degradation. The mechanistic details behind the fate of HIV-1 Envelope are lacking. Since the substantive claim of the manuscript is the distinctive mechanism(s) by which MARCH8 exhibits its antiviral effects on HIV-1 and VSV – it is necessary that at least the following comment be addressed:The loss of cell surface localization of HIV-1 Envelope upon MARCH8 overexpression is striking. However, as presented, the details regarding intracellular retention of HIV-1 Env are not clear. This is an essential claim of this study and thus it is recommended that microscopy experiments be performed to elucidate this mechanism: where is HIV-1 Env being sorted? This can be easily done by performing colocalization staining with known intracellular compartment markers.

We appreciate the reviewer’s comments. As suggested, we performed colocalization staining by using an antibody against the trans-Golgi network (TGN) marker TGN46, and found that in the presence of MARCH8, HIV-1 Env preferentially colocalized with TGN46, suggesting that MARCH8 can sequester HIV-1 Env in the TGN without degradation, as we previously hypothesized. We have added the data in Figure 1F, and the related text in the first paragraph of the Results and Discussion.

Overall, the mechanism by which MARCH-8 internalizes HIV Env is under-developed. The observation that the tyrosine at position 222 is required is interesting. It is tempting to speculate that one or more AP complexes are involved, but the authors state their preliminary data, which they do not show, does not support this hypothesis. Despite this, in their model they still speculate that AP complexes are involved. The contradictory statements are very confusing.

We thank the reviewer for this point. We apologize for the confusion caused by our statements concerning the possible binding of MARCH8’s YxxΦ motif with AP μ-subunits, which sound contradictory. The reason why we could not see the interaction between AP μ-subunits and MARCH8 in our preliminary data was not based on the results obtained under established conditions, but rather based on those due to the technical difficulties in detecting specific bands in our IP/Western blotting assays. We have therefore amended the related text in the last paragraph of the Results and Discussion.

The authors also need to explain how the levels of MARCH8 expression levels in their system compare to endogenous levels in HIV target cells. Over expression studies are notorious for artifacts and if the levels are higher, this must be noted.

In our previous paper (Tada et al., 2015), by performing real-time RT-PCR, we optimized DNA amounts of MARCH8 expression plasmid (120 ng/a total of 1 μg) for transfection into 293T cells, based on the endogenous mRNA levels of expression in macrophages that express higher levels of MARCH8 mRNA. Here in this study, we used an aforementioned optimized amount of MARCH8 plasmid for transfection of 293T cells throughout the experiments. We have added the related text in the Materials and methods section (subsection “Viral infectivity assays”).